# A Note on the Reverse Order Law for *g*-Inverse of Operator Product

Yingying Qin and Zhiping Xiong *

School of Mathematics and Computational Science, Wuyi University, Jiangmen 529020, China; qyy@163.com
* Correspondence: xzpwhere@163.com

**Abstract:** The generalized inverse has many important applications in aspects of the theoretical research of matrices and statistics. One of the core problems of the generalized inverse is finding the necessary and sufficient conditions of the reverse order laws for the generalized inverse of the operator product. In this paper, we study the reverse order law for the *g*-inverse of an operator product $T_1 T_2 T_3$ using the technique of matrix form of bounded linear operators. In particular, some necessary and sufficient conditions for the inclusion $T_3\{1\}T_2\{1\}T_1\{1\} \subseteq (T_1T_2T_3)\{1\}$ is presented. Moreover, some finite dimensional results are extended to infinite dimensional settings.

**Keywords:** *g*-inverse; reverse order law; bounded linear operator; operator product; Hilbert space

**MSC:** 47A05; 15A09; 15A24



## 1. Introduction

Throughout this paper 'an operator' means 'a bounded linear operator over Hilbert space'. $\mathbb{H}$, $\mathbb{I}$, $\mathbb{J}$ and $\mathbb{K}$ denote arbitrary Hilbert spaces. $L(\mathbb{H}, \mathbb{K})$ denote the set of all bounded linear operators from $\mathbb{H}$ to $\mathbb{K}$ and $L(\mathbb{H}) = \mathbb{L}(\mathbb{H}, \mathbb{H})$. $I$ denotes the identity operator over Hilbert space and $O$ is the zero operator over Hilbert space. For an operator $T \in L(\mathbb{H}, \mathbb{K})$, $T^*$, $R(T)$ and $N(T)$ denote the adjoint operator, the range and the null-space of $T$, respectively.

Recall that, an operator $X \in L(\mathbb{K}, \mathbb{H})$ is called the Moore–Penrose inverse of $T \in L(\mathbb{H}, \mathbb{K})$, if $X$ satisfies the following four operator equations [1]:

$$(1)\ TXT = T,\ \ (2)\ XTX = X,\ \ (3)\ (TX)^* = TX,\ \ (4)\ (XT)^* = XT. \tag{1}$$

If such operator $X$ exists, then it is unique and is denoted by $T^\dagger$. It is well known that the Moore–Penrose inverse of $T$ exists if, and only if, $R(T)$ is closed, see [2,3].

For any operator $T \in L(\mathbb{H}, \mathbb{K})$, let $T\{i, j, \cdots, k\}$ denote the set of operators $X \in L(\mathbb{K}, \mathbb{H})$ which satisfy equations $(i), (j), \cdots, (k)$ from among Equations (1)–(4) of Formula (1). An operator in $T\{i, j, \cdots, k\}$ is called an $\{i, j, \cdots, k\}$-inverse of $T$ and denoted by $T^{(i,j,\cdots,k)}$. For example, an operator $X$ of the set $T\{1\}$ is called a $\{1\}$-inverse or a *g*-inverse of $T$ and is denoted by $T^{(1)}$. The well-known seven common types of generalized inverse of $T$ introduced from (1.1) are, respectively, the $\{1\}$-inverse, $\{1,2\}$-inverse, $\{1,3\}$-inverse, $\{1,4\}$-inverse, $\{1,2,3\}$-inverse, $\{1,2,4\}$-inverse and $\{1,2,3,4\}$-inverse, the last being the unique Moore–Penrose inverse. In particular, when $T$ is nonsingular, then it is easily seen that $T^\dagger = T^{-1}$. We refer the reader to [2–4] for basic results on generalized inverses.

The concepts of the generalized inverse were shown to be very useful in various applied mathematical settings. For example, applications to singular differential or difference equations, Markov chains, cryptography, iterative method or multibody system dynamics, and so on, which can be found in [2,3,5–8]. In the above applied mathematical settings, large-scale scientific computing problems eventually translate to least square problems. Using generalized inverse to give some fast and effective iterative solution algorithms for

these least square problems has attracted considerable attention, and many interesting results have been obtained, see [2,3,9–11].

Suppose $T_i$, $i = 1, 2, 3$, and $\beta$ are bounded linear operators over Hilbert space, the least squares problem is finding $x$ that minimizes the norm:

$$\min_x \|(T_1 T_2 T_3)x - \beta\|_2, \tag{2}$$

which is used in many practical scientific problems. Any solution $x$ of the above LS can be expressed as $x = (T_1 T_2 T_3)^{(1,3)}\beta$. If $(T_1 T_2 T_3)x = \beta$ is consistent, the minimum norm solution $x$ has the form $x = (T_1 T_2 T_3)^{(1,4)}\beta$. The unique minimal norm least square solution $x$ of the above LS is $x = (T_1 T_2 T_3)^{\dagger}\beta$. One of the problems concerning the above LS is under what conditions the reverse order law

$$T_3^{(1,j,...,k)} T_2^{(1,j,...,k)} T_1^{(1,j,...,k)} \subseteq (T_1 T_2 T_3)^{(1,j,...,k)} \tag{3}$$

holds.

If Formula (3) is true, then, according to the reverse order law Formula (3) and the iterative algorithm theory, we can naturally construct some ideal iterative sequence, and then design some fast and effective iterative algorithms to solve Formula (2). If Formula (3) is not necessarily true, can we find the necessary and sufficient conditions for Formula (3)? Applying the reverse order law to design some fast and effective iterative algorithms to solve Formula (2), will avoid multiple decompositions of the correlation matrices and keep it in each iteration. The structure of the iterative sequence reduces the amount of machine storage, maintains the convergence, stability of the algorithm, and improves the operation speed, see [2,3,9,11–13].

The reverse order law for generalized inverses of operator (or matrix) product yields a class of interesting problems that are fundamental in the theory of generalized inverses, see [2,3], which have attracted considerable attention since the middle 1960s, and many interesting results have been obtained, see [14–22].

For the generalized inverses of matrix product, Greville [7] first gave a necessary and sufficient condition for $(AB)^{\dagger} \subseteq B^{\dagger}A^{\dagger}$. Since then, the problem of the reverse order law for generalized inverses of a matrix product was studied widely. Hartwig [8] derived the necessary and sufficient conditions for the Moore–Penrose inverse of the product of three matrices, and Y. Tian [14] obtained the reverse order law for the Moore–Penrose inverse of the products of multiple matrices. M. Wei [15] and De Pierro [16], respectively, derived necessary and sufficient conditions for the reverse order laws $B\{1\}A\{1\} \subseteq (AB)\{1\}$ and $B\{1,2\}A\{1,2\} \subseteq (AB)\{1,2\}$, by applying product singular value decomposition (PSVD). M. Wei [17] then deduced necessary and sufficient conditions for reverse order laws for $g$-inverse of multiple matrix product. For $A_n\{1,2,k\}A_{n-1}\{1,2,k\}\cdots A_1\{1,2,k\} \subseteq (A_1 A_2 \cdots A_n)\{1,2,k\}$, $k = 3, 4$, Xiong and Zheng [18] presented the equivalent conditions using extremal ranks of the generalized Schur complement.

For the generalized inverses of operator product, Bouldin [5] and Izumino [19] extended the results of Greville [7] to the bounded linear operators on Hilbert space, using the gaps between subspaces. Let $T_1 \in L(\mathbb{H}, \mathbb{L})$ and $T_2 \in L(\mathbb{K}, \mathbb{H})$, such that the product $T_1 T_2$ is meaningful, using the technique of matrix form of bounded linear operators, D.S.Djordjević [20] showed that the reverse order law $T_2^{\dagger}T_1^{\dagger} = (T_1 T_2)^{\dagger}$ holds if, and only if, $R(T_1^* T_1 T_2) \subseteq R(T_2)$ and $R(T_2 T_2^* T_1^*) \subseteq R(T_1^*)$. J.Kohila et al. [21] obtained the necessary and sufficient conditions for the reverse order law of the Moore–Penrose inverse in rings with involutions. In [22], D.S.Cvetković-Ilić et al., studied this reverse order law of the Moore–Penrose inverse in $C^*$-algebra. The reader can find more results of the reverse order law for the Moore–Penrose inverse of operator product in [23–27].

Recently, Xiong and Qin [28,29] studied the reverse order laws for $\{1,3\}$−inverse and $\{1,4\}$−inverse of operator products, using the technique of matrix form of bounded linear operators [30] and some equivalent conditions are derived for these reverse order laws. With the same threads of [28,29], in this paper, we will study the reverse order law for

the *g*-inverse of an operator product $T_1 T_2 T_3$. In particular, some necessary and sufficient conditions for the reverse order law

$$T_3\{1\}T_2\{1\}T_1\{1\} \subseteq (T_1 T_2 T_3)\{1\} \tag{4}$$

is presented. Moreover, some finite dimensional results are extended to infinite dimensional settings.

## 2. A Set of Lemmas

As the main tools of our discussion, we first present the following lemmas.

**Lemma 1** ([30])**.** *Suppose $T \in L(\mathbb{H}, \mathbb{K})$ that have a closed range. Let $H_1$ and $H_2$ be closed and mutually orthogonal subspaces of $\mathbb{H}$, such that $H_1 \oplus H_2 = \mathbb{H}$, and $K_1$, $K_2$ be closed and mutually orthogonal subspaces of $\mathbb{K}$, such that $\mathbb{K} = K_1 \oplus K_2$. Then the operator $T$ has the following matrix representations with respect to the orthogonal sums of subspaces $\mathbb{H} = H_1 \oplus H_2 = R(T^*) \oplus N(T)$ and $\mathbb{K} = K_1 \oplus K_2 = R(T) \oplus N(T^*)$:*

(1) $T = \begin{pmatrix} T_{11} & T_{12} \\ O & O \end{pmatrix} : \begin{pmatrix} H_1 \\ H_2 \end{pmatrix} \to \begin{pmatrix} R(T) \\ N(T^*) \end{pmatrix}$ *and* $T^\dagger = \begin{pmatrix} T_{11}^* E^{-1} & O \\ T_{12}^* E^{-1} & O \end{pmatrix} : \begin{pmatrix} R(T) \\ N(T^*) \end{pmatrix} \to \begin{pmatrix} H_1 \\ H_2 \end{pmatrix}$,
   *where $E = T_{11} T_{11}^* + T_{12} T_{12}^*$ is invertible on $R(T)$;*

(2) $T = \begin{pmatrix} T_{11} & O \\ T_{21} & O \end{pmatrix} : \begin{pmatrix} R(T^*) \\ N(T) \end{pmatrix} \to \begin{pmatrix} K_1 \\ K_2 \end{pmatrix}$ *and* $T^\dagger = \begin{pmatrix} F^{-1} T_{11}^* & F^{-1} T_{12}^* \\ O & O \end{pmatrix} : \begin{pmatrix} K_1 \\ K_2 \end{pmatrix} \to \begin{pmatrix} R(T^*) \\ N(T) \end{pmatrix}$,
   *where $F = T_{11}^* T_{11} + T_{21}^* T_{21}$ is invertible on $R(T^*)$;*

(3) $T = \begin{pmatrix} T_{11} & O \\ O & O \end{pmatrix} : \begin{pmatrix} R(T^*) \\ N(T) \end{pmatrix} \to \begin{pmatrix} R(T) \\ N(T^*) \end{pmatrix}$ *and* $T^\dagger = \begin{pmatrix} T_{11}^{-1} & O \\ O & O \end{pmatrix} : \begin{pmatrix} R(T) \\ N(T^*) \end{pmatrix} \to \begin{pmatrix} R(T^*) \\ N(T) \end{pmatrix}$,
   *where $T_{11}$ is invertible on $R(T^*)$.*

**Lemma 2** ([2,3])**.** *Let $T \in L(\mathbb{H}, \mathbb{K})$ have a closed range and $G \in L(\mathbb{K}, \mathbb{H})$. Then the following statements are equivalent:*

(1) $TGT = T \Leftrightarrow G \in T\{1\}$;

(2) *there exists some $X \in L(\mathbb{K}, \mathbb{H})$, such that $G = T^\dagger + X - T^\dagger T X T T^\dagger$.*

**Lemma 3** ([2,3])**.** *Let $T \in L(\mathbb{H}, \mathbb{K})$ and $N \in L(\mathbb{K}, \mathbb{H})$ have closed ranges. Then*

(1) $TT^\dagger N = N \Leftrightarrow R(N) \subseteq R(T)$;

(2) $NT^\dagger T = N \Leftrightarrow R(N^*) \subseteq R(T^*)$.

**Lemma 4** ([24])**.** *Let $T \in L(\mathbb{H}, \mathbb{K})$ and $N \in L(\mathbb{M}, \mathbb{N})$ have closed ranges. Then $TXN = O$ for every $X \in L(\mathbb{N}, \mathbb{H})$ if and only if $T = O$ or $N = O$.*

## 3. Main Results

Let $T_1 \in L(\mathbb{J}, \mathbb{K})$, $T_2 \in L(\mathbb{I}, \mathbb{J})$ and $T_3 \in L(\mathbb{H}, \mathbb{I})$ where $T_1$, $T_2$, $T_3$ and $T_1 T_2 T_3$ are regular operators. From Lemma 1, we know that the operators $T_1$, $T_2$ and $T_3$ have the following matrix forms with respect to the orthogonal sum of subspaces:

$$T_1 = \begin{pmatrix} T_1^{11} & T_1^{12} \\ O & O \end{pmatrix} : \begin{pmatrix} R(T_2) \\ N(T_2^*) \end{pmatrix} \to \begin{pmatrix} R(T_1) \\ N(T_1^*) \end{pmatrix}, \tag{5}$$

$$T_1^\dagger = \begin{pmatrix} (T_1^{11})^* D^{-1} & O \\ (T_1^{12})^* D^{-1} & O \end{pmatrix} : \begin{pmatrix} R(T_1) \\ N(T_1^*) \end{pmatrix} \to \begin{pmatrix} R(T_2) \\ N(T_2^*) \end{pmatrix}, \tag{6}$$

where $D = T_1^{11}(T_1^{11})^* + T_1^{12}(T_1^{12})^*$ is invertible on $R(T_1)$.

$$T_2 = \begin{pmatrix} T_2^{11} & O \\ O & O \end{pmatrix} : \begin{pmatrix} R(T_2^*) \\ N(T_2) \end{pmatrix} \to \begin{pmatrix} R(T_2) \\ N(T_2^*) \end{pmatrix}, \tag{7}$$

$$T_2^\dagger = \begin{pmatrix} (T_2^{11})^{-1} & O \\ O & O \end{pmatrix} : \begin{pmatrix} R(T_2) \\ N(T_2^*) \end{pmatrix} \to \begin{pmatrix} R(T_2^*) \\ N(T_2) \end{pmatrix}, \tag{8}$$

where $T_2^{11}$ is invertible on $R(T_2^*)$.

$$T_3 = \begin{pmatrix} T_3^{11} & O \\ T_3^{21} & O \end{pmatrix} : \begin{pmatrix} R(T_3^*) \\ N(T_3) \end{pmatrix} \to \begin{pmatrix} R(T_2^*) \\ N(T_2) \end{pmatrix}, \tag{9}$$

$$T_3^\dagger = \begin{pmatrix} S^{-1}(T_3^{11})^* & S^{-1}(T_3^{21})^* \\ O & O \end{pmatrix} : \begin{pmatrix} R(T_2^*) \\ N(T_2) \end{pmatrix} \to \begin{pmatrix} R(T_3^*) \\ N(T_3) \end{pmatrix}, \tag{10}$$

where $S = (T_3^{11})^* T_3^{11} + (T_3^{21})^* T_3^{21}$ is invertible on $R(T_3^*)$.

According to the Formulas (5)–(10), we have the following theorem.

**Theorem 1.** *Let* $T_1 \in L(\mathbb{J}, \mathbb{K})$, $T_2 \in L(\mathbb{I}, \mathbb{J})$ *and* $T_3 \in L(\mathbb{H}, \mathbb{I})$ *where* $T_1$, $T_2$ *and* $T_3$ *are regular operators. Then*

(1) $N(T_1) \subseteq R(T_2) \Leftrightarrow (T_1^{12})^* D^{-1} T_1^{11} = O$ *and* $(T_1^{12})^* D^{-1} T_1^{12} = I$;

(2) $N(T_2) \subseteq R(T_3) \Leftrightarrow T_3^{21} S^{-1}(T_3^{11})^* = O$ *and* $T_3^{21} S^{-1}(T_3^{21})^* = I$;

(3) $R(T_2^\dagger(I - T_1^\dagger T_1)) \subseteq R(T_3) \Leftrightarrow (I - T_3^{11} S^{-1}(T_3^{11})^*)(T_2^{11})^{-1}(I - (T_1^{11})^* D^{-1} T_1^{11}) = O$ *and*

$$\begin{aligned} (I - T_3^{11} S^{-1}(T_3^{11})^*)(T_2^{11})^{-1}(T_1^{11})^* D^{-1} T_1^{12} &= O \text{ and} \\ - T_3^{21} S^{-1}(T_3^{11})^*(T_2^{11})^{-1}(I - (T_1^{11})^* D^{-1} T_1^{11}) &= O \text{ and} \\ - T_3^{11} S^{-1}(T_3^{11})^*(T_2^{11})^{-1}(T_1^{11})^* D^{-1} T_1^{12} &= O. \end{aligned}$$

**Proof.** By Lemma 3, we know that

$$N(T_1) \subseteq R(T_2) \Leftrightarrow (I - T_2 T_2^\dagger)(I - T_1^\dagger T_1) = O, \tag{11}$$

$$N(T_2) \subseteq R(T_3) \Leftrightarrow (I - T_3 T_3^\dagger)(I - T_2^\dagger T_2) = O, \tag{12}$$

and

$$R(T_2^\dagger(I - T_1^\dagger T_1)) \subseteq R(T_3) \Leftrightarrow (I - T_3 T_3^\dagger) T_2^\dagger(I - T_1^\dagger T_1) = O. \tag{13}$$

Combining the Formulas (5)–(10) with the Formulas (11)–(13), we have

$$\begin{aligned} N(T_1) \subseteq R(T_2) \quad &\Leftrightarrow \quad (I - T_2 T_2^\dagger)(I - T_1^\dagger T_1) = O \\ &\Leftrightarrow \quad (T_1^{12})^* D^{-1} T_1^{11} = O \text{ and } (T_1^{12})^* D^{-1} T_1^{12} = I, \end{aligned} \tag{14}$$

$$\begin{aligned} N(T_2) \subseteq R(T_3) \quad &\Leftrightarrow \quad (I - T_3 T_3^\dagger)(I - T_2^\dagger T_2) = O \\ &\Leftrightarrow \quad T_3^{21} S^{-1}(T_3^{11})^* = O \text{ and } T_3^{21} S^{-1}(T_3^{21})^* = I \end{aligned} \tag{15}$$

and

$$
\begin{aligned}
R(T_2^\dagger (I - T_1^\dagger T_1)) \subseteq R(T_3) \quad \Leftrightarrow \quad & (I - T_3 T_3^\dagger) T_2^\dagger (I - T_1^\dagger T_1) = O \\
\Leftrightarrow \quad & (I - T_3^{11} S^{-1} (T_3^{11})^*)(T_2^{11})^{-1}(I - (T_1^{11})^* D^{-1} T_1^{11}) = O, \\
& (I - T_3^{11} S^{-1} (T_3^{11})^*)(T_2^{11})^{-1}(T_1^{11})^* D^{-1} T_1^{12} = O, \\
& -T_3^{21} S^{-1} (T_3^{11})^* (T_2^{11})^{-1}(I - (T_1^{11})^* D^{-1} T_1^{11}) = O, \\
& -T_3^{11} S^{-1} (T_3^{11})^* (T_2^{11})^{-1}(T_1^{11})^* D^{-1} T_1^{12} = O.
\end{aligned}
\tag{16}
$$

From (14)–(16), we have Theorem 1. □

From Lemma 1, we know that the operator $T_1 T_2 T_3$ has the following matrix forms with respect to the orthogonal sum of subspaces:

$$
T_1 T_2 T_3 = \begin{pmatrix} T_1^{11} T_2^{11} T_3^{11} & O \\ O & O \end{pmatrix} : \begin{pmatrix} R(T_3^*) \\ N(T_3) \end{pmatrix} \to \begin{pmatrix} R(T_1) \\ N(T_1^*) \end{pmatrix}
\tag{17}
$$

and

$$
(T_1 T_2 T_3)^* = \begin{pmatrix} (T_3^{11})^* (T_2^{11})^* (T_1^{11})^* & O \\ O & O \end{pmatrix} : \begin{pmatrix} R(T_1) \\ N(T_1^*) \end{pmatrix} \to \begin{pmatrix} R(T_3^*) \\ N(T_3) \end{pmatrix}.
\tag{18}
$$

Combining (5)–(10) with the results in Lemma 2, it follows that there exist three bounded linear operators $W, H, P$:

$$
W = \begin{pmatrix} W_{11} & W_{12} \\ W_{21} & W_{22} \end{pmatrix} : \begin{pmatrix} R(T_1) \\ N(T_1^*) \end{pmatrix} \to \begin{pmatrix} R(T_2) \\ N(T_2^*) \end{pmatrix},
$$

where $W \in L(\mathbb{K}, \mathbb{J})$ and $W_{11}, W_{12}, W_{21}, W_{22}$ are arbitrary bounded linear operators on appropriate spaces.

$$
H = \begin{pmatrix} H_{11} & H_{12} \\ H_{21} & H_{22} \end{pmatrix} : \begin{pmatrix} R(T_2) \\ N(T_2^*) \end{pmatrix} \to \begin{pmatrix} R(T_2^*) \\ N(T_2) \end{pmatrix},
$$

where $H \in L(\mathbb{J}, \mathbb{I})$ and $H_{11}, H_{12}, H_{21}, H_{22}$ are arbitrary bounded linear operators on appropriate spaces.

$$
P = \begin{pmatrix} P_{11} & P_{12} \\ P_{21} & P_{22} \end{pmatrix} : \begin{pmatrix} R(T_2^*) \\ N(T_2) \end{pmatrix} \to \begin{pmatrix} R(T_3^*) \\ N(T_3) \end{pmatrix},
$$

where $P \in L(\mathbb{I}, \mathbb{H})$ and $P_{11}, P_{12}, P_{21}, P_{22}$ are arbitrary bounded linear operators on appropriate spaces.

Furthermore, by Lemma 2, we have

$$
T_1^{(1)} = T_1^\dagger + W - T_1^\dagger T_1 W T_1 T_1^\dagger = \begin{pmatrix} \tau_{11} & W_{12} \\ \tau_{21} & W_{22} \end{pmatrix}, \; where
\tag{19}
$$

$$
\begin{aligned}
\tau_{11} &= (T_1^{11})^* D^{-1} + W_{11} - (T_1^{11})^* D^{-1} T_1^{11} W_{11} - (T_1^{11})^* D^{-1} T_1^{12} W_{21}, \\
\tau_{21} &= (T_1^{12})^* D^{-1} + W_{21} - (T_1^{12})^* D^{-1} T_1^{12} W_{21} - (T_1^{12})^* D^{-1} T_1^{11} W_{11}.
\end{aligned}
$$

$$
T_2^{(1)} = T_2^\dagger + H - T_2^\dagger T_2 H T_2 T_2^\dagger = \begin{pmatrix} (T_2^{11})^{-1} & H_{12} \\ H_{21} & H_{22} \end{pmatrix},
\tag{20}
$$

$$
T_3^{(1)} = T_3^\dagger + P - T_3^\dagger T_3 P T_3 T_3^\dagger = \begin{pmatrix} \mu_{11} & \mu_{12} \\ P_{21} & P_{22} \end{pmatrix}, \; where
\tag{21}
$$

$$\mu_{11} = S^{-1}(T_3^{11})^* + P_{11} - P_{11}T_3^{11}S^{-1}(T_3^{11})^* - P_{12}T_3^{21}S^{-1}(T_3^{11})^*,$$
$$\mu_{12} = S^{-1}(T_3^{21})^* + P_{12} - P_{12}T_3^{21}S^{-1}(T_3^{21})^* - P_{11}T_3^{11}S^{-1}(T_3^{21})^*.$$

Combining the Formulas (17)–(21) with the results in Theorem 1, we obtain the main result of this paper.

**Theorem 2.** *Let $T_1 \in L(\mathbb{J}, \mathbb{K})$, $T_2 \in L(\mathbb{I}, \mathbb{J})$ and $T_3 \in L(\mathbb{H}, \mathbb{I})$, where $T_1$, $T_2$, $T_3$ and $T_1T_2T_3$ are regular operators. Then the following statements are equivalent:*

(1) $T_3\{1\}T_2\{1\}T_1\{1\} \subseteq (T_1T_2T_3)\{1\}$;

(2) $N(T_1) \subseteq R(T_2)$, $N(T_2) \subseteq R(T_3)$ *and* $R(T_2^\dagger(I - T_1^\dagger T_1)) \subseteq R(T_3)$.

**Proof.** From the Formula 1 in Lemma 1, we know that the reverse order law (4) holds, i.e., the conditions 1 in Theorem 2 holds, if, and only if,

$$(T_1T_2T_3)T_3^{(1)}T_2^{(1)}T_1^{(1)}(T_1T_2T_3) = T_1T_2T_3 \tag{22}$$

holds for any $T_i^{(1)} \in T_i\{1\}$, $i = 1, 2, 3$.

By (17) and (19)–(21), we have

$$
\begin{aligned}
&(T_1T_2T_3)T_3^{(1)}T_2^{(1)}T_1^{(1)}(T_1T_2T_3) \\
&= \begin{pmatrix} T_1^{11}T_2^{11}T_3^{11} & O \\ O & O \end{pmatrix} \times \begin{pmatrix} \mu_{11} & \mu_{12} \\ P_{21} & P_{22} \end{pmatrix} \times \begin{pmatrix} (T_2^{11})^{-1} & H_{12} \\ H_{21} & H_{22} \end{pmatrix} \times \begin{pmatrix} \tau_{11} & W_{12} \\ \tau_{21} & W_{22} \end{pmatrix} \\
&\quad \times \begin{pmatrix} T_1^{11}T_2^{11}T_3^{11} & O \\ O & O \end{pmatrix} \\
&= \begin{pmatrix} \nu_{11} & \nu_{12} \\ O & O \end{pmatrix} \begin{pmatrix} T_1^{11}T_2^{11}T_3^{11} & O \\ O & O \end{pmatrix} = \begin{pmatrix} \nu_{11}T_1^{11}T_2^{11}T_3^{11} & O \\ O & O \end{pmatrix},
\end{aligned} \tag{23}
$$

where

$$
\begin{aligned}
\nu_{11} &= (T_1^{11}T_2^{11}T_3^{11}\mu_{11}(T_2^{11})^{-1})\tau_{11} + (T_1^{11}T_2^{11}T_3^{11}\mu_{12}H_{22})\tau_{11} \\
&\quad + (T_1^{11}T_2^{11}T_3^{11}\mu_{11}(T_2^{11})^{-1})\tau_{21} + (T_1^{11}T_2^{11}T_3^{11}\mu_{12}H_{22})\tau_{21},
\end{aligned}
$$

$$
\begin{aligned}
\nu_{12} &= (T_1^{11}T_2^{11}T_3^{11}\mu_{11}(T_2^{11})^{-1})W_{12} + (T_1^{11}T_2^{11}T_3^{11}\mu_{12}H_{21})W_{12} \\
&\quad + (T_1^{11}T_2^{11}T_3^{11}\mu_{11}H_{12})W_{22} + (T_1^{11}T_2^{11}T_3^{11}\mu_{12}H_{22})W_{22}.
\end{aligned}
$$

In the rest of this section, we will prove that the conditions 1 in Theorem 2 is equal to the conditions 2 in Theorem 2, i.e., the Formula (22) is equal to the Formulas (14)–(16) in Theorem 1.

*Conditions*2 $\Rightarrow$ *Conditions*1: Combining (23) with (14)–(16), we have

$$\nu_{11}T_1^{11}T_2^{11}T_3^{11} = T_1^{11}T_2^{11}T_3^{11}$$

and

$$
\begin{aligned}
&(T_1T_2T_3)T_3^{(1)}T_2^{(1)}T_1^{(1)}(T_1T_2T_3) \\
&= \begin{pmatrix} T_1^{11}T_2^{11}T_3^{11} & O \\ O & O \end{pmatrix} \times \begin{pmatrix} \mu_{11} & \mu_{12} \\ P_{21} & P_{22} \end{pmatrix} \times \begin{pmatrix} (T_2^{11})^{-1} & H_{12} \\ H_{21} & H_{22} \end{pmatrix} \times \begin{pmatrix} \tau_{11} & W_{12} \\ \tau_{21} & W_{22} \end{pmatrix} \\
&\quad \times \begin{pmatrix} T_1^{11}T_2^{11}T_3^{11} & O \\ O & O \end{pmatrix} \\
&= \begin{pmatrix} \nu_{11}T_1^{11}T_2^{11}T_3^{11} & O \\ O & O \end{pmatrix} = \begin{pmatrix} T_1^{11}T_2^{11}T_3^{11} & O \\ O & O \end{pmatrix} \\
&= T_1T_2T_3. \tag{24}
\end{aligned}
$$

That is, from (14)–(16), we have (22), i.e., we proved that *Conditions*2 $\Rightarrow$ *Conditions*1.

*Conditions*1 $\Rightarrow$ *Conditions*2: If the reverse order law $T_3\{1\}T_2\{1\}T_1\{1\} \subseteq (T_1T_2T_3)\{1\}$ holds. From Lemma 2, we known that the equation $(T_1T_2T_3)T_3^{(1)}T_2^{(1)}T_1^{(1)}(T_1T_2T_3) = T_1T_2T_3$ holds for any $T_i^{(1)} \in T_i\{1\}$, $i = 1, 2, 3$.

Firstly, let

$$T_1^{(1)} = T_1^\dagger + (I - T_1^\dagger T_1)W = \begin{pmatrix} \eta_{11} & \eta_{12} \\ \eta_{21} & \eta_{22} \end{pmatrix}, \tag{25}$$

where

$$
\begin{aligned}
\eta_{11} &= (T_1^{11})^* D^{-1} + (I - (T_1^{11})^* D^{-1} T_1^{11})W_{11} - (T_1^{11})^* D^{-1} T_1^{12} W_{21}, \\
\eta_{12} &= (I - (T_1^{11})^* D^{-1} T_1^{11})W_{12} - (T_1^{11})^* D^{-1} T_1^{12} W_{22}, \\
\eta_{21} &= (T_1^{12})^* D^{-1} + (I - (T_1^{12})^* D^{-1} T_1^{12})W_{21} - (T_1^{12})^* D^{-1} T_1^{11} W_{11}, \\
\eta_{22} &= (I - (T_1^{12})^* D^{-1} T_1^{12})W_{22} - (T_1^{12})^* D^{-1} T_1^{11} W_{12},
\end{aligned}
$$

$$T_2^{(1)} = T_2^\dagger = \begin{pmatrix} (T_2^{11})^{-1} & O \\ O & O \end{pmatrix}, \tag{26}$$

$$T_3^{(1)} = T_3^\dagger + P(I - T_3 T_3^\dagger) = \begin{pmatrix} \rho_{11} & \rho_{12} \\ \rho_{21} & \rho_{22} \end{pmatrix}, \tag{27}$$

where

$$
\begin{aligned}
\rho_{11} &= S^{-1}(T_3^{11})^* + P_{11}(I - T_3^{11} S^{-1}(T_3^{11})^*) - P_{12} T_3^{21} S^{-1}(T_3^{11})^*, \\
\rho_{12} &= S^{-1}(T_3^{11})^* + P_{12}(I - T_3^{21} S^{-1}(T_3^{21})^*) - P_{11} T_3^{11} S^{-1}(T_3^{21})^*, \\
\rho_{21} &= P_{21}(I - T_3^{11} S^{-1}(T_3^{11})^*) - P_{22} T_3^{21} S^{-1}(T_3^{11})^*, \\
\rho_{22} &= P_{22}(I - T_3^{21} S^{-1}(T_3^{21})^*) - P_{21} T_3^{11} S^{-1}(T_3^{21})^*.
\end{aligned}
$$

Then, we have

$$(T_1T_2T_3)T_3^{(1)}T_2^\dagger T_1^{(1)}(T_1T_2T_3) = T_1T_2T_3 \tag{28}$$

and

$$
\begin{aligned}
&(T_1T_2T_3)T_3^{(1)}T_2^\dagger T_1^{(1)}(T_1T_2T_3) = T_1T_2T_3 \\
\Leftrightarrow\ & \begin{pmatrix} T_1^{11} T_2^{11} T_3^{11} & O \\ O & O \end{pmatrix} \times \begin{pmatrix} \rho_{11} & \rho_{12} \\ \rho_{21} & \rho_{22} \end{pmatrix} \times \begin{pmatrix} (T_2^{11})^{-1} & O \\ O & O \end{pmatrix} \times \begin{pmatrix} \eta_{11} & \eta_{12} \\ \eta_{21} & \eta_{22} \end{pmatrix} \times \begin{pmatrix} T_1^{11} T_2^{11} T_3^{11} & O \\ O & O \end{pmatrix} \\
=\ & \begin{pmatrix} T_1^{11} T_2^{11} T_3^{11} & O \\ O & O \end{pmatrix}.
\end{aligned} \tag{29}
$$

Since $W_{ij}$ ($i, j = 1, 2$) are arbitrary, let $W_{11} = W_{12} = W_{21} = W_{22} = O$, we have

$$
\begin{aligned}
&(T_1T_2T_3)T_3^{(1)}T_2^\dagger T_1^{(1)}(T_1T_2T_3) = T_1T_2T_3 \\
\Leftrightarrow\ & \begin{pmatrix} T_1^{11} T_2^{11} T_3^{11} & O \\ O & O \end{pmatrix} \times \begin{pmatrix} \rho_{11} & \rho_{12} \\ \rho_{21} & \rho_{22} \end{pmatrix} \times \begin{pmatrix} (T_2^{11})^{-1} & O \\ O & O \end{pmatrix} \times \begin{pmatrix} (T_1^{11})^* D^{-1} & O \\ (T_1^{12})^* D^{-1} & O \end{pmatrix} \\
& \times \begin{pmatrix} T_1^{11} T_2^{11} T_3^{11} & O \\ O & O \end{pmatrix} = \begin{pmatrix} T_1^{11} T_2^{11} T_3^{11} & O \\ O & O \end{pmatrix}.
\end{aligned} \tag{30}
$$

Combining (29) with (30), we have

$$
\begin{pmatrix} T_1^{11}T_2^{11}T_3^{11} & O \\ O & O \end{pmatrix} \times \begin{pmatrix} \rho_{11} & \rho_{12} \\ \rho_{21} & \rho_{22} \end{pmatrix} \times \begin{pmatrix} (T_2^{11})^{-1} & O \\ O & O \end{pmatrix} \times \begin{pmatrix} \eta_{11} - (T_1^{11})^* D^{-1} & \eta_{12} \\ \eta_{21} - (T_1^{12})^* D^{-1} & \eta_{22} \end{pmatrix}
$$
$$
\times \begin{pmatrix} T_1^{11}T_2^{11}T_3^{11} & O \\ O & O \end{pmatrix} = \begin{pmatrix} O & O \\ O & O \end{pmatrix}. \tag{31}
$$

Let $P_{11} = P_{12} = P_{21} = P_{22} = O$, then from (31), we have

$$
\begin{pmatrix} T_1^{11}T_2^{11}T_3^{11} & O \\ O & O \end{pmatrix} \times \begin{pmatrix} S^{-1}(T_3^{11})^* & S^{-1}(T_3^{21})^* \\ O & O \end{pmatrix} \times \begin{pmatrix} (T_2^{11})^{-1} & O \\ O & O \end{pmatrix} \times \begin{pmatrix} \eta_{11} - (T_1^{11})^* D^{-1} & \eta_2 \\ \eta_{21} - (T_1^{12})^* D^{-1} & \eta_{22} \end{pmatrix}
$$
$$
\times \begin{pmatrix} T_1^{11}T_2^{11}T_3^{11} & O \\ O & O \end{pmatrix} = \begin{pmatrix} O & O \\ O & O \end{pmatrix}. \tag{32}
$$

From (31) and (32), we get

$$
\begin{pmatrix} T_1^{11}T_2^{11}T_3^{11} & O \\ O & O \end{pmatrix} \times \begin{pmatrix} \rho_{11} - S^{-1}(T_3^{11})^* & \rho_{12} - S^{-1}(T_3^{21})^* \\ \rho_{21} & \rho_{22} \end{pmatrix} \times \begin{pmatrix} (T_2^{11})^{-1} & O \\ O & O \end{pmatrix}
$$
$$
\times \begin{pmatrix} (\eta_{11} - (T_1^{11})^* D^{-1} & \eta_2 \\ \eta_{21} - (T_1^{12})^* D^{-1} & \eta_{22} \end{pmatrix} \times \begin{pmatrix} T_1^{11}T_2^{11}T_3^{11} & O \\ O & O \end{pmatrix} = \begin{pmatrix} O & O \\ O & O \end{pmatrix}. \tag{33}
$$

According to the Equation (33), we have

$$
\begin{aligned}
& T_1^{11}T_2^{11}T_3^{11} P_{11}(I - T_3^{11}S^{-1}(T_3^{11})^*)(T_2^{11})^{-1}(I - (T_1^{11})^* D^{-1}T_1^{11})W_{11}T_1^{11}T_2^{11}T_3^{11} \\
& - T_1^{11}T_2^{11}T_3^{11} P_{12}T_3^{21}S^{-1}(T_3^{11})^*(T_2^{11})^{-1}(I - (T_1^{11})^* D^{-1}T_1^{11})W_{11}T_1^{11}T_2^{11}T_3^{11} \\
& - T_1^{11}T_2^{11}T_3^{11} P_{11}(I - T_3^{11}S^{-1}(T_3^{11})^*)(T_2^{11})^{-1}(T_1^{11})^* D^{-1}T_1^{12}W_{21}T_1^{11}T_2^{11}T_3^{11} \\
& + T_1^{11}T_2^{11}T_3^{11} P_{12}T_3^{21}S^{-1}(T_3^{11})^*(T_2^{11})^{-1}(T_1^{11})^* D^{-1}T_1^{12}W_{21}T_1^{11}T_2^{11}T_3^{11} \\
& = O.
\end{aligned} \tag{34}
$$

Since $P_{11}$, $P_{12}$ and $W_{11}$, $W_{21}$ are arbitrary, then from Lemma 4 and (34), we have

$$
\begin{aligned}
(I - T_3^{11}S^{-1}(T_3^{11})^*)(T_2^{11})^{-1}(I - (T_1^{11})^* D^{-1}T_1^{11}) &= O, \\
(I - T_3^{11}S^{-1}(T_3^{11})^*)(T_2^{11})^{-1}(T_1^{11})^* D^{-1}T_1^{12} &= O, \\
-T_3^{21}S^{-1}(T_3^{11})^*(T_2^{11})^{-1}(I - (T_1^{11})^* D^{-1}T_1^{11}) &= O, \\
-T_3^{11}S^{-1}(T_3^{11})^*(T_2^{11})^{-1}(T_1^{11})^* D^{-1}T_1^{12} &= O.
\end{aligned} \tag{35}
$$

From (35) and (16), we get that if the reverse order law $T_3\{1\}T_2\{1\}T_1\{1\} \subseteq (T_1T_2T_3)\{1\}$ holds, then $R(T_2^\dagger(I - T_1^\dagger T_1)) \subseteq R(T_3)$.

Secondly, let

$$
T_1^{(1)} = T_1^\dagger = \begin{pmatrix} (T_1^{11})^* D^{-1} & O \\ (T_1^{12})^* D^{-1} & O \end{pmatrix}, \tag{36}
$$

$$
T_2^{(1)} = T_2^\dagger + (I - T_2 T_2^\dagger)H = \begin{pmatrix} (T_2^{11})^{-1} & O \\ H_{21} & H_{22} \end{pmatrix}, \tag{37}
$$

$$
T_3^{(1)} = T_3^\dagger + P(I - T_3 T_3^\dagger) = \begin{pmatrix} \rho_{11} & \rho_{12} \\ \rho_{21} & \rho_{22} \end{pmatrix}, \tag{38}
$$

where

$$
\begin{aligned}
\rho_{11} &= S^{-1}(T_3^{11})^* + P_{11}(I - T_3^{11}S^{-1}(T_3^{11})^*) - P_{12}T_3^{21}S^{-1}(T_3^{11})^*,\\
\rho_{12} &= S^{-1}(T_3^{11})^* + P_{12}(I - T_3^{21}S^{-1}(T_3^{21})^*) - P_{11}T_3^{11}S^{-1}(T_3^{21})^*,\\
\rho_{21} &= P_{21}(I - T_3^{11}S^{-1}(T_3^{11})^*) - P_{22}T_3^{21}S^{-1}(T_3^{11})^*,\\
\rho_{22} &= P_{22}(I - T_3^{21}S^{-1}(T_3^{21})^*) - P_{21}T_3^{11}S^{-1}(T_3^{21})^*.
\end{aligned}
$$

Then, we have

$$
(T_1 T_2 T_3) T_3^{(1)} T_2^{(1)} T_1^\dagger (T_1 T_2 T_3) = T_1 T_2 T_3 \tag{39}
$$

and

$$
\begin{aligned}
&(T_1 T_2 T_3) T_3^{(1)} T_2^{(1)} T_1^\dagger (T_1 T_2 T_3) = T_1 T_2 T_3\\
\Leftrightarrow\ &\begin{pmatrix} T_1^{11} T_2^{11} T_3^{11} & O \\ O & O \end{pmatrix} \times \begin{pmatrix} \rho_{11} & \rho_{12} \\ \rho_{21} & \rho_{22} \end{pmatrix} \times \begin{pmatrix} (T_2^{11})^{-1} & O \\ H_{21} & H_{22} \end{pmatrix} \times \begin{pmatrix} (T_1^{11})^* D^{-1} & O \\ (T_1^{12})^* D^{-1} & O \end{pmatrix}\\
&\times \begin{pmatrix} T_1^{11} T_2^{11} T_3^{11} & O \\ O & O \end{pmatrix} = \begin{pmatrix} T_1^{11} T_2^{11} T_3^{11} & O \\ O & O \end{pmatrix}.
\end{aligned} \tag{40}
$$

Since $H_{ij}$ $(i, j = 1, 2)$ are arbitrary, let $H_{21} = H_{22} = O$, we have

$$
\begin{aligned}
&(T_1 T_2 T_3) T_3^{(1)} T_2^{(1)} T_1^\dagger (T_1 T_2 T_3) = T_1 T_2 T_3\\
\Leftrightarrow\ &\begin{pmatrix} T_1^{11} T_2^{11} T_3^{11} & O \\ O & O \end{pmatrix} \times \begin{pmatrix} \rho_{11} & \rho_{12} \\ \rho_{21} & \rho_{22} \end{pmatrix} \times \begin{pmatrix} (T_2^{11})^{-1} & O \\ O & O \end{pmatrix} \times \begin{pmatrix} (T_1^{11})^* D^{-1} & O \\ (T_1^{12})^* D^{-1} & O \end{pmatrix}\\
&\times \begin{pmatrix} T_1^{11} T_2^{11} T_3^{11} & O \\ O & O \end{pmatrix} = \begin{pmatrix} T_1^{11} T_2^{11} T_3^{11} & O \\ O & O \end{pmatrix}.
\end{aligned} \tag{41}
$$

Combining (40) with (41), we have

$$
\begin{aligned}
&\begin{pmatrix} T_1^{11} T_2^{11} T_3^{11} & O \\ O & O \end{pmatrix} \times \begin{pmatrix} \rho_{11} & \rho_{12} \\ \rho_{21} & \rho_{22} \end{pmatrix} \times \begin{pmatrix} O & O \\ H_{21} & H_{22} \end{pmatrix} \times \begin{pmatrix} (T_1^{11})^* D^{-1} & O \\ (T_1^{12})^* D^{-1} & O \end{pmatrix}\\
&\times \begin{pmatrix} T_1^{11} T_2^{11} T_3^{11} & O \\ O & O \end{pmatrix} = \begin{pmatrix} O & O \\ O & O \end{pmatrix}.
\end{aligned} \tag{42}
$$

Let $P_{11} = P_{12} = P_{21} = P_{22} = O$, then from (42), we have

$$
\begin{aligned}
&\begin{pmatrix} T_1^{11} T_2^{11} T_3^{11} & O \\ O & O \end{pmatrix} \times \begin{pmatrix} S^{-1}(T_3^{11})^* & S^{-1}(T_3^{21})^* \\ O & O \end{pmatrix} \times \begin{pmatrix} O & O \\ H_{21} & H_{22} \end{pmatrix} \times \begin{pmatrix} (T_1^{11})^* D^{-1} & O \\ (T_1^{12})^* D^{-1} & O \end{pmatrix}\\
&\times \begin{pmatrix} T_1^{11} T_2^{11} T_3^{11} & O \\ O & O \end{pmatrix} = \begin{pmatrix} O & O \\ O & O \end{pmatrix}.
\end{aligned} \tag{43}
$$

From (42) and (43), we get

$$
\begin{aligned}
&\begin{pmatrix} T_1^{11} T_2^{11} T_3^{11} & O \\ O & O \end{pmatrix} \times \begin{pmatrix} \rho_{11} - S^{-1}(T_3^{11})^* & \rho_{12} - S^{-1}(T_3^{21})^* \\ \rho_{21} & \rho_{22} \end{pmatrix} \times \begin{pmatrix} O & O \\ H_{21} & H_{22} \end{pmatrix}\\
&\times \begin{pmatrix} (T_1^{11})^* D^{-1} & O \\ (T_1^{12})^* D^{-1} & O \end{pmatrix} \times \begin{pmatrix} T_1^{11} T_2^{11} T_3^{11} & O \\ O & O \end{pmatrix} = \begin{pmatrix} O & O \\ O & O \end{pmatrix}.
\end{aligned} \tag{44}
$$

According to the Equation (44), we have

$$
\begin{aligned}
&T_1^{11} T_2^{11} T_3^{11} P_{12}(I - T_3^{21}S^{-1}(T_3^{21})^*) H_{21}(T_1^{11})^* D^{-1} T_1^{11} T_2^{11} T_3^{11}\\
&+ T_1^{11} T_2^{11} T_3^{11} P_{12}(I - T_3^{21}S^{-1}(T_3^{21})^*) H_{22}(T_1^{12})^* D^{-1} T_1^{11} T_2^{11} T_3^{11}\\
&- T_1^{11} T_2^{11} T_3^{11} P_{11} T_3^{11} S^{-1}(T_3^{21})^* H_{21}(T_1^{11})^* D^{-1} T_1^{11} T_2^{11} T_3^{11}\\
&- T_1^{11} T_2^{11} T_3^{11} P_{11} T_3^{11} S^{-1}(T_3^{21})^* H_{22}(T_1^{12})^* D^{-1} T_1^{11} T_2^{11} T_3^{11} = O.
\end{aligned} \tag{45}
$$

Since $P_{11}$, $P_{12}$ and $H_{21}$, $H_2 2$ are arbitrary, then from Lemma 4 and (45), we have

$$I - T_3^{21} S^{-1}(T_3^{21})^* = O \text{ and } T_3^{11} S^{-1}(T_3^{21})^* = O. \tag{46}$$

From (15) and (46), we get that if the reverse order law $T_3\{1\}T_2\{1\}T_1\{1\} \subseteq (T_1 T_2 T_3)\{1\}$ holds, then $N(T_2) \subseteq R(T_3)$.

Thirdly, let

$$T_1^{(1)} = T_1^\dagger + (I - T_1^\dagger T_1)W = \begin{pmatrix} \eta_{11} & \eta_{12} \\ \eta_{21} & \eta_{22} \end{pmatrix}, \tag{47}$$

where

$$
\begin{aligned}
\eta_{11} &= (T_1^{11})^* D^{-1} + (I - (T_1^{11})^* D^{-1} T_1^{11})W_{11} - (T_1^{11})^* D^{-1} T_1^{12} W_{21}, \\
\eta_{12} &= (I - (T_1^{11})^* D^{-1} T_1^{11})W_{12} - (T_1^{11})^* D^{-1} T_1^{12} W_{22}, \\
\eta_{21} &= (T_1^{12})^* D^{-1} + (I - (T_1^{12})^* D^{-1} T_1^{12})W_{21} - (T_1^{12})^* D^{-1} T_1^{11} W_{11}, \\
\eta_{22} &= (I - (T_1^{12})^* D^{-1} T_1^{12})W_{22} - (T_1^{12})^* D^{-1} T_1^{11} W_{12},
\end{aligned}
$$

$$T_2^{(1)} = T_2^\dagger + H(I - T_2 T_2^\dagger) = \begin{pmatrix} (T_2^{11})^{-1} & H_{12} \\ O & H_{22} \end{pmatrix}, \tag{48}$$

$$T_3^{(1)} = T_3^\dagger = \begin{pmatrix} S^{-1}(T_3^{11})^* & S^{-1}(T_3^{21})^* \\ O & O \end{pmatrix}. \tag{49}$$

Then we have

$$(T_1 T_2 T_3) T_3^\dagger T_2^{(1)} T_1^{(1)} (T_1 T_2 T_3) = T_1 T_2 T_3 \tag{50}$$

and

$$
\begin{aligned}
& (T_1 T_2 T_3) T_3^\dagger T_2^{(1)} T_1^{(1)} (T_1 T_2 T_3) = T_1 T_2 T_3 \\
\Leftrightarrow\ & \begin{pmatrix} T_1^{11} T_2^{11} T_3^{11} & O \\ O & O \end{pmatrix} \times \begin{pmatrix} S^{-1}(T_3^{11})^* & S^{-1}(T_3^{21})^* \\ O & O \end{pmatrix} \times \begin{pmatrix} (T_2^{11})^{-1} & H_{12} \\ O & H_{22} \end{pmatrix} \times \begin{pmatrix} \eta_{11} & \eta_{12} \\ \eta_{21} & \eta_{22} \end{pmatrix} \\
& \times \begin{pmatrix} T_1^{11} T_2^{11} T_3^{11} & O \\ O & O \end{pmatrix} = \begin{pmatrix} T_1^{11} T_2^{11} T_3^{11} & O \\ O & O \end{pmatrix}.
\end{aligned} \tag{51}
$$

Since $W_{ij}$ ($i, j = 1, 2$) are arbitrary, let $W_{11} = W_{12} = W_{21} = W_{22} = O$, we have

$$
\begin{aligned}
& (T_1 T_2 T_3) T_3^{(1)} T_2^\dagger T_1^{(1)} (T_1 T_2 T_3) = T_1 T_2 T_3 \\
\Leftrightarrow\ & \begin{pmatrix} T_1^{11} T_2^{11} T_3^{11} & O \\ O & O \end{pmatrix} \times \begin{pmatrix} S^{-1}(T_3^{11})^* & S^{-1}(T_3^{21})^* \\ O & O \end{pmatrix} \times \begin{pmatrix} (T_2^{11})^{-1} & H_{12} \\ O & H_{22} \end{pmatrix} \times \begin{pmatrix} (T_1^{11})^* D^{-1} & O \\ (T_1^{12})^* D^{-1} & O \end{pmatrix} \\
& \times \begin{pmatrix} T_1^{11} T_2^{11} T_3^{11} & O \\ O & O \end{pmatrix} = \begin{pmatrix} T_1^{11} T_2^{11} T_3^{11} & O \\ O & O \end{pmatrix}.
\end{aligned} \tag{52}
$$

Combining (51) with (52), we have

$$
\begin{aligned}
& \begin{pmatrix} T_1^{11} T_2^{11} T_3^{11} & O \\ O & O \end{pmatrix} \times \begin{pmatrix} S^{-1}(T_3^{11})^* & S^{-1}(T_3^{21})^* \\ O & O \end{pmatrix} \times \begin{pmatrix} (T_2^{11})^{-1} & H_{12} \\ O & H_{22} \end{pmatrix} \\
& \times \begin{pmatrix} \eta_{11} - (T_1^{11})^* D^{-1} & \eta_{12} \\ \eta_{21} - (T_1^{12})^* D^{-1} & \eta_{22} \end{pmatrix} \times \begin{pmatrix} T_1^{11} T_2^{11} T_3^{11} & O \\ O & O \end{pmatrix} = \begin{pmatrix} O & O \\ O & O \end{pmatrix}.
\end{aligned} \tag{53}
$$

Let $H_{12} = H_{22} = O$, then from (53) we have

$$
\begin{pmatrix} T_1^{11} T_2^{11} T_3^{11} & O \\ O & O \end{pmatrix} \times \begin{pmatrix} S^{-1}(T_3^{11})^* & S^{-1}(T_3^{21})^* \\ O & O \end{pmatrix} \times \begin{pmatrix} (T_2^{11})^{-1} & O \\ O & O \end{pmatrix}
$$
$$
\times \begin{pmatrix} \eta_{11} - (T_1^{11})^* D^{-1} & \eta_2 \\ \eta_{21} - (T_1^{12})^* D^{-1} & \eta_{22} \end{pmatrix} \times \begin{pmatrix} T_1^{11} T_2^{11} T_3^{11} & O \\ O & O \end{pmatrix} = \begin{pmatrix} O & O \\ O & O \end{pmatrix}. \tag{54}
$$

From (53) and (54), we get

$$
\begin{pmatrix} T_1^{11} T_2^{11} T_3^{11} & O \\ O & O \end{pmatrix} \times \begin{pmatrix} S^{-1}(T_3^{11})^* & S^{-1}(T_3^{21})^* \\ O & O \end{pmatrix} \times \begin{pmatrix} O & H_{12} \\ O & H_{22} \end{pmatrix}
$$
$$
\times \begin{pmatrix} \eta_{11} - (T_1^{11})^* D^{-1} & \eta_2 \\ \eta_{21} - (T_1^{12})^* D^{-1} & \eta_{22} \end{pmatrix} \times \begin{pmatrix} T_1^{11} T_2^{11} T_3^{11} & O \\ O & O \end{pmatrix} = \begin{pmatrix} O & O \\ O & O \end{pmatrix}. \tag{55}
$$

According to the Formula (55), we have

$$
\begin{aligned}
& T_1^{11} T_2^{11} T_3^{11} S^{-1}(T_3^{11})^* H_{12}(I - (T_1^{12})^* D^{-1} T_1^{12}) W_{21} T_1^{11} T_2^{11} T_3^{11} \\
& + T_1^{11} T_2^{11} T_3^{11} S^{-1}(T_3^{21})^* H_{22}(I - (T_1^{12})^* D^{-1} T_1^{12}) W_{21} T_1^{11} T_2^{11} T_3^{11} \\
& - T_1^{11} T_2^{11} T_3^{11} S^{-1}(T_3^{11})^* H_{12}(T_1^{12})^* D^{-1} T_1^{11} W_{11} T_1^{11} T_2^{11} T_3^{11} \\
& - T_1^{11} T_2^{11} T_3^{11} S^{-1}(T_3^{21})^* H_{22}(T_1^{12})^* D^{-1} T_1^{11} W_{11} T_1^{11} T_2^{11} T_3^{11} \\
& = O.
\end{aligned} \tag{56}
$$

Since $H_{12}$, $H_{22}$ and $W_{11}$, $W_{21}$ are arbitrary, then from Lemma 4 and (56), we have

$$
I - (T_1^{12})^* D^{-1} T_1^{12} = O \text{ and } (T_1^{12})^* D^{-1} T_1^{11} = O. \tag{57}
$$

From (14) and (57), we get that if the reverse order law $T_3\{1\} T_2\{1\} T_1\{1\} \subseteq (T_1 T_2 T_3)\{1\}$ holds, then $N(T_1) \subseteq R(T_2)$.

Finally, combining (35), (46) with (57), we prove that if the reverse order law Formula (4) holds, the equalities (14)–(16) also hold. That is, *Conditions*1 $\Rightarrow$ *Conditions*2.　□

**Corollary 1.** *Let $T_1 \in L(\mathbb{J}, \mathbb{K})$ and $T_2 \in L(\mathbb{I}, \mathbb{J})$, where $T_1$, $T_2$ and $T_1 T_2$ are regular operators. Then the following statements are equivalent:*

(1) $T_2\{1\} T_1\{1\} \subseteq (T_1 T_2)\{1\}$;

(2) $N(T_1) \subseteq R(T_2)$.

## 4. Conclusions

Many problems in applied sciences, such as non-linear control theory, matrix analysis, statistics and numerical linear algebra are closely related to the least squares problems of the operator equation $(T_1 T_2 \cdots T_n)x = b$. If the operator equation is consistent, any solution $x$ of the above equation can be expressed as $x = (T_1 T_2 \cdots T_n)^{(1)} b$. One of the problems concerning the above least squares problems is under what conditions the reverse order law for the *g*-inverse of operator product holds. In this paper, by using the technique of matrix form of bounded linear operators, we study the reverse order law for the *g*-inverse of an operator product $T_1 T_2 T_3$. In particular, some necessary and sufficient conditions for the inclusion $T_3\{1\} T_2\{1\} T_1\{1\} \subseteq (T_1 T_2 T_3)\{1\}$ is presented. Moreover, some finite dimensional results are extended to infinite dimensional settings. The work in this paper provides a useful tool in many algorithms for the computation of the least squares solutions of operator equations.

**Author Contributions:** All authors have contributed equally to this work. All authors have read and agreed to the published version of the manuscript.

**Funding:** This work was supported by the Project for Characteristic Innovation of 2018 Guangdong University (No: 2018KTSCX234), the Natural Science Foundation of Guangdong (No: 2014A030313625), the Basic Theory and Scientific Research of Science and Technology Project of Jiangmen City, China (No: 2020JC01010, 2021030102610005049) and the Joint Research and Development Fund of Wuyi University, Hong Kong and Macao (No: 2019WGALH20).

**Institutional Review Board Statement:** Studies not involving humans or animals.

**Informed Consent Statement:** Not applicable.

**Data Availability Statement:** Not applicable.

**Acknowledgments:** The authors would like to thank Varvari Radu and the anonymous referees for their very detailed comments and constructive suggestions, which greatly improved the presentation of this paper.

**Conflicts of Interest:** The authors declare no conflict of interest.

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
