# Peer review of "A Note on the Reverse Order Law for g-Inverse of Operator Product"

_axioms, doi:10.3390/axioms11050226_

Round 1

Reviewer 1 Report

Report on the paper Axioms 1625935
Title: A note on the reverse order law for g-inverse of operator product
Authors: Yingying Qin, Zhiping Xiong
In this paper, the authors consider the reverse order law for g-inverse of the product of three
operators. There is only one theorem in the paper. The method of the proof is elementary and
based on the well-know computational techniques. There is no motivation for such kind of
result neither serious applications. For all these reasons I cannot recommend this paper for
publication in an as highly ranking journal as Axioms.

Author Response

Revision  Letter

Dear Sir,

We have received your letter dated 18-Mar-2022, giving us your advices for revision

of the manuscript (axioms-1625935). We appreciate very much for your comments

and valuable suggestions. The manuscript has been carefully revised as the comments

and the response listed as followed.

Reviewer's comment (1):  Moderate English changes required.
  Reply: We have revised according to the reviewer's suggestion in revised MS. The English language and style have been carefully checked and revised

Reviewer's comment (2): There is only one theorem in the paper. The method of the proof is elementary and based on the well-know computational techniques. There is no motivation for such kind of result neither serious applications.   

Reply: In recent years, the study of operators play a vital rule in many areas of mathematics and physics. In this paper, we studied the reverse order law for g-inverse of three operator products by using the technique of matrix form of bounded linear operators. The work in this paper is a useful tool in algorithms for the computation of the least squares solution of operator equations. The research method of this paper is the extension and application of the existing method. The research ideas and results of our paper are novel and concise than the conclusions of the existing literature.

The reviewer gave us many concrete suggestions to improve the manuscript. We thank them very much for the positive and constructive comments to the MS in the ACKNOWLEDGMENTS.

Thank you very much!

Sincerely yours

Dr. Zhiping Xiong

School of Mathematics and Computational Science, Wuyi University, Jiangmen

529020, P. R. China

Reviewer 2 Report

The proposed manuscript is focused on the reverse order law for g-inverse of operator product. The investigated issues adhere to the scientific scope of the Journal. The main editorial requirements are also satisfied.

The analyzed problem is worth investigating, however the paper needs substantial revision. The structure of the paper needs improvement. The text lacks sufficient Introduction section - there is no literature review provided, no literature gap defined, no aim and scope of the paper presented.

The Introduction section in the presented form should be renamed and given as the modelling section.

The paper lacks any conclusion section. There is also a lack of a brief discussion on the possibility of implementing the results obtained in practice. 

Author Response

Revision  Letter

Dear Sir,

We have received your letter dated 18-Mar-2022, giving us your advices for revision

of the manuscript (axioms-1625935). We appreciate very much for your comments

and valuable suggestions. The manuscript has been carefully revised as the comments

and the response listed as followed.

Reviewer's comment (1):  English language and style are fine/minor spell check required.

Reply: We have revised according to the reviewer's suggestion in revised MS. The English language and style have been carefully checked and revised.

Reviewer's comment (2): The analyzed problem is worth investigating, however the paper needs substantial revision. The structure of the paper needs improvement. The text lacks sufficient Introduction section - there is no literature review provided, no literature gap defined, no aim and scope of the paper presented. The Introduction section in the presented form should be renamed and given as the modelling section.

Reply: We have revised according to the reviewer's suggestion in revised MS. See the Introduction section of revised MS in pages 1-3.

Reviewer's comment (3): The paper lacks any conclusion section. There is also a lack of a brief discussion on the possibility of implementing the results obtained in practice.

Reply: We have revised according to the reviewer's suggestion in revised MS. See the Conclusion section of revised MS in page 13.

The reviewers gave us many concrete suggestions to improve the manuscript. We thank them very much for the positive and constructive comments to the MS in the ACKNOWLEDGMENTS.

Thank you very much!

Sincerely yours

Dr. Zhiping Xiong

School of Mathematics and Computational Science, Wuyi University, Jiangmen

529020, P. R. China

Reviewer 3 Report

In recent years the study of operators play a vital rule in many areas of mathematics and physics. In this paper the reverse order law for g-inverse of three operators’ product by using the technique of matrix form of bounded linear operators have been studied. 

The paper can be accepted but only after some major revision. First of all, the authors should correct the abstract section. Also the abstract section is too short. 

Secondly, I'm particularly concerned with the “Introduction" as the goals and objectives are scattered all over that section. During a first reading of the manuscript, it is not clear what the authors are trying to convey nor the paper's objectives are clearly identified. Thus, to improve the manuscript's clarity, I would suggest modifying the Introduction.

Thirdly, the author should give the lemmas in the new section.

My fourth Recommendation is, in the revised paper, the authors should include a new section. “Concluding Remarks and Observations”. Particularly, the novelty and relevance of the newly introduced classes should be emphasized, pointing out their potential applications and perspective for future research. Such information can be found throughout the manuscript, but it must be particularly highlighted in the conclusions.  

Finally the authors should check the article thoroughly for spelling and typo mistakes. The author should also know the correct usage of the comma and full-stops. In many places author have miss used these pronunciations.

Author Response

Revision  Letter

Dear Sir,

We have received your letter dated 18-Mar-2022, giving us your advices for revision

of the manuscript (axioms-1625935). We appreciate very much for your comments

and valuable suggestions. The manuscript has been carefully revised as the comments

and the response listed as followed.

Reviewer's comment (1): Moderate English changes required.

Reply: We have revised according to the reviewer's suggestion in revised MS. The English language and style have been carefully checked and revised.

Reviewer's comment (2): The paper can be accepted but only after some major revision. First of all, the authors should correct the abstract section. Also the abstract section is too short. 

Reply: We have revised according to the reviewer's suggestion in revised MS. See the Abstract section of revised MS in page 1.

Reviewer's comment (3): Secondly, I'm particularly concerned with the “Introduction" as the goals and objectives are scattered all over that section. During a first reading of the manuscript, it is not clear what the authors are trying to convey nor the paper's objectives are clearly identified. Thus, to improve the manuscript's clarity, I would suggest modifying the Introduction.

Reply: We have revised according to the reviewer's suggestion in revised MS. See the Introduction section of revised MS in pages 1-3.

Reviewer's comment (4): Thirdly, the author should give the lemmas in the new section.

Reply: We have revised according to the reviewer's suggestion in revised MS. See revised, page 3, lines 27-38 and page 4, lines 1-9.

Reviewer's comment (5): My fourth Recommendation is, in the revised paper, the authors should include a new section. “Concluding Remarks and Observations”. Particularly, the novelty and relevance of the newly introduced classes should be emphasized, pointing out their potential applications and perspective for future research. Such information can be found throughout the manuscript, but it must be particularly highlighted in the conclusions.  

Reply: We have revised according to the reviewer's suggestion in revised MS. See the Conclusion section of revised MS in page 13.

Reviewer's comment (6): Finally the authors should check the article thoroughly for spelling and typo mistakes. The author should also know the correct usage of the comma and full-stops. In many places author have miss used these pronunciations.

Reply: We have revised according to the reviewer's suggestion in revised MS. We carefully and thoroughly checked the article for spelling and misspellings.

The reviewers gave us many concrete suggestions to improve the manuscript. We thank them very much for the positive and constructive comments to the MS in the ACKNOWLEDGMENTS.

Thank you very much!

Sincerely yours

Dr. Zhiping Xiong

School of Mathematics and Computational Science, Wuyi University, Jiangmen

529020, P. R. China

Round 2

Reviewer 1 Report

I still have the same opinion:

The method of the proof is elementary and based on the well-know computational techniques.  There is no motivation for such kind of result neither serious applications. For all these reasons I cannot recommend this paper for publication in an as highly ranking journal as Axioms.

Author Response

Dear Sir,

We have received your letter dated 04-Apr-2022, giving us your advices for revision of the manuscript (axioms-1625935). We appreciate very much for your comments and valuable suggestions. The manuscript has been carefully revised as the comments and the response listed as followed.

Reviewer's comment (1):  English language and style are fine/minor spell check required.
Reply: We have revised according to the reviewer's suggestion in revised MS. The English language and style have been carefully checked and revised

Reviewer's comment (2): The method of the proof is elementary and based on the well-know computational techniques. There is no motivation for such kind of result neither serious applications.   

Reply: In recent years, the study of operators play a vital rule in many areas of mathematics and physics. The work in this paper is a useful tool in algorithms for the computation of the least squares solution of operator equations. The research method of this paper is the extension and application of the well-know computational techniques. The research ideas and results of our paper are novel and concise than the conclusions of the existing literature.

You gave us many concrete suggestions to improve the manuscript. We thank you very much for the positive and constructive comments to the MS in the ACKNOWLEDGMENTS.

Thank you very much!

Sincerely yours

Dr. Zhiping Xiong

School of Mathematics and Computational Science, Wuyi University, Jiangmen

529020, P. R. China

Reviewer 2 Report

The proposed manuscript has been improved according to the reviewer's suggestions. However, the paper stil lacks literature gap defined, the aim and scope is missing either. The implementation possibilities description should be provided.

Author Response

Dear Sir,

We have received your letter dated 04-Apr-2022, giving us your advices for revision of the manuscript (axioms-1625935). We appreciate very much for your comments and valuable suggestions. The manuscript has been carefully revised as the comments and the response listed as followed.

Reviewer's comment (1):  English language and style are fine/minor spell check required.

Reply: We have revised according to the reviewer's suggestion in revised MS. The English language and style have been carefully checked and revised.

Reviewer's comment (2): The paper still lacks literature gap defined, the aim and scope is missing either. The implementation possibilities description should be provided.

Reply: We have revised according to the reviewer's suggestion in revised MS. See the Introduction section of revised MS in pages 1-3 and the Conclusion section of revised MS in page 13.

You gave us many concrete suggestions to improve the manuscript. We thank you very much for the positive and constructive comments to the MS in the ACKNOWLEDGMENTS.

Thank you very much!

Sincerely yours

Dr. Zhiping Xiong

School of Mathematics and Computational Science, Wuyi University, Jiangmen

529020, P. R. China

Reviewer 3 Report

The paper is improved significantly, therefore I recommend its publication in *Axioms*.  

One point: The Lammas should be include in a new section 

Section "A Set of Lemmas"

Author Response

Dear Sir,

We have received your letter dated 04-Apr-2022, giving us your advices for revision of the manuscript (axioms-1625935). We appreciate very much for your comments and valuable suggestions. The manuscript has been carefully revised as the comments and the response listed as followed.

Reviewer's comment (1): English language and style are fine/minor spell check required.

Reply: We have revised according to the reviewer's suggestion in revised MS. The English language and style have been carefully checked and revised.

Reviewer's comment (2): The Lammas should be included in a new section. Section "A Set of Lemmas".

Reply: We have revised according to the reviewer's suggestion in revised MS. See the Section of revised MS in page 3.

You gave us many concrete suggestions to improve the manuscript. We thank you very much for the positive and constructive comments to the MS in the ACKNOWLEDGMENTS.

Thank you very much!

Sincerely yours

Dr. Zhiping Xiong

School of Mathematics and Computational Science, Wuyi University, Jiangmen

529020, P. R. China
